# Comparative Efficacy of Classic Versus Horizontal Incision Techniques in Skin-Reducing Mastectomy: A Single Center Retrospective Analysis

**DOI:** 10.3390/jcm13206276

**Published:** 2024-10-21

**Authors:** Andrea Vittorio Emanuele Lisa, Alessandro Mela, Sergio Miranda, Mario Alessandri Bonetti, Manuela Bottoni, Mattia Intra, Eleonora Pagan, Vincenzo Bagnardi, Mario Rietjens

**Affiliations:** 1Department of Plastic and Reconstructive Surgery, European Institute of Oncology, IRCCS, 20141 Milan, Italy; alessandromela3@gmail.com (A.M.); sergio.miranda@ieo.it (S.M.); mario.alessandribonetti@ieo.it (M.A.B.); manuela.bottoni@ieo.it (M.B.); mattia.intra@ieo.it (M.I.); mario.rietjens@ieo.it (M.R.); 2PhD Program in Applied Medical-Surgical Sciences, Department of Surgical Sciences, University of Rome “Tor Vergata”, Viale Oxford 81, 00133 Rome, Italy; 3Humanitas University Department of Biomedical Sciences, Humanitas University, Via Rita Levi Montalcini 4, Pieve Emanuele, 20090 Milan, Italy; 4Division of Breast Surgery, European Institute of Oncology, Istituto di Ricovero e Cura a Carattere Scientifico (IRCCS), 20141 Milan, Italy; 5Department of Statistics and Quantitative Methods, University of Milano-Bicocca, 20126 Milan, Italy; eleonora.pagan@unimib.it (E.P.); vincenzo.bagnardi@ieo.it (V.B.)

**Keywords:** mastectomy, skin reducing mastectomy, breast, breast reconstruction, implant, horizontal incision, vertical incision, voluminous breast

## Abstract

**Background**: The reconstruction of large breasts carries a heightened risk profile. While skin-reducing mastectomy (SRM) techniques facilitate the correction of breast ptosis, they are frequently associated with a high incidence of vascular complications. This study compares two SRM techniques—the horizontal incision and the classic inverted T incision—by examining their clinical and surgical outcomes. **Methods**: We retrospectively analyzed data from 24 patients (30 breasts) who underwent SRM with immediate prosthetic reconstruction between 2019 and 2023 at the European Institute of Oncology in Milan, Italy. Our comparison focused on breast aesthetic outcome, reconstruction quality, complication rates (early and late), and patient satisfaction, utilizing the BREAST-Q questionnaire to gauge the latter. **Results**: Among the 24 patients included in the study, 16 (20 breasts) were treated with the inverted T technique, and 8 (10 breasts) with the horizontal incision approach. A higher overall complication rate was observed with the inverted T technique compared to the horizontal method, with early complications outnumbering late ones. The most common issues were recurrent seroma and skin necrosis leading to implant exposure. Notably, there were no cases of implant infection. Although the horizontal incision technique achieved slightly higher patient satisfaction scores, the difference was not statistically significant. **Discussion**: The inverted T and horizontal incision techniques each have unique benefits and drawbacks. Our findings indicate enhanced patient satisfaction and reduced complication rates with the horizontal incision technique. The selection of the technique should be customized based on the patient’s individual risk factors, tissue quality, and preferences.

## 1. Introduction

In recent decades, due to advancements in breast screening and early detection of cancer, there has been a steady increase in the number of patients undergoing mastectomy for breast cancer treatment [1]. Consequently, the demand for breast reconstruction surgeries has escalated. Patients with large breasts pose a considerable challenge for plastic surgeons due to the heightened morbidity, increased risk of surgical complications, contributory risk factors, and compromised tissue quality [2]. In response to these challenges, innovative surgical techniques have been developed for the reconstruction of significantly large breasts. Among them, the skin-reducing mastectomy (SRM) technique has gained progressive prominence. It effectively reduces excess skin after mastectomy, providing a more anatomical breast shape and facilitating the creation of a suitable prosthetic pocket for either an implant or a tissue expander. Despite its benefits in achieving good symmetry with the contralateral breast and favorable cosmetic outcomes, SRM is associated with a higher incidence of complications, particularly concerning the viability of the mastectomy flaps [3]. To mitigate such risks, a horizontal incision skin reduction technique has been introduced by Kontos et al. [4], which significantly lessens the likelihood of vascularization impairments. However, this method could result in a more conspicuous scar and potential asymmetry with the contralateral breast especially in case of inverted T mastopexy or reduction. Given the lack of comparative studies between these two SRM techniques, our research aims to evaluate the differences in clinical and surgical outcomes, assess aesthetic quality, and determine complication rates, patient satisfaction, and healing times.

## 2. Materials and Methods

We extracted information from our prospectively collected institutional database on all consecutive patients who underwent a skin-reducing mastectomy at the European Institute of Oncology (EIO) in Milan (Italy), between 2019 and 2023. The inclusion criteria for participating patients encompassed a significant breast weight, the employment of a horizontal or inverted T incision technique, immediate reconstruction using definitive implants, and completion of follow-up. Exclusions were made for reconstructions with tissue expanders and loss to follow-up.

The choice of inverted T or horizontal incision technique was based upon different patients’ features. According to the lower rate of complications and the greater preservation of vascular supply, the horizontal incision technique may be more suitable for patients with:Advanced disease: This includes cases with advanced stages or more extensive involvement of surrounding tissues.Compromised nipple-areola complex vascularity: This can occur due to factors like previous surgery or radiation therapy and smoking habits.Lower tissue quality: Factors such as smoking, diabetes, or previous surgery can negatively impact tissue quality.Elevated risk of complications: Comorbidities like diabetes, obesity, or cardiovascular disease can increase the likelihood of complications.

Furthermore, the horizontal incision technique may be preferable for patients opting for immediate implant-based reconstruction. The absence of a T-junction in this technique potentially reduces the risk of dehiscence, promoting better wound healing and aesthetic outcomes.

However, further research with a larger sample size and prospective design is necessary to confirm these observations and establish definitive recommendations.

Patients who underwent the procedure with a horizontal incision were matched 1:2 with patients who underwent inverted T skin-reducing mastectomy. The matching variables were age, BMI, breast prosthesis volume and bilateral mastectomy. Patient satisfaction was evaluated using the BREAST-Q questionnaire. Early and late complications together with reconstruction failure described as breast implant explants were recorded during follow-up. Informed consent was obtained from all subjects involved in the study. This study was conducted in accordance with the Declaration of Helsinki and approved by the Data Governance Board of EIO (protocol code UID 4548 approved on 8 April 2024).

### 2.1. Surgical Techniques

-Horizontal incision skin-reducing technique [4]: This technique incorporates a horizontal incision centered around the nipple-areola complex (NAC), followed by de-epithelialization of the tissue within the incision while preserving the NAC (Figure 1). Only in one case the nipple was grafted. Mastectomy is accessed through a classical radial incision. All dermal layers are conserved and folded upon themselves. After mastectomy, a pocket for implant placement is created through the same incision, bounded superiorly by the detached pectoralis major muscle at its inferior and inferomedial attachments. If the inferior pole of the mastectomy flap is insufficiently thick, an acellular dermal matrix is applied and secured to the margin of the pectoralis major muscle and the fascia at the inframammary fold. The de-epithelialized dermal flap is then folded upon itself, aligning and suturing the incision edges. Although a horizontal incision is made, the skin reduction vector is vertical, which mitigates the risk of NAC misalignment upon repositioning (Figure 2).-Inverted T skin-reducing technique: As described by Nava [5], this technique involves 11 cm vertical incisions to facilitate lower skin trimming, with the mastectomy incision parallel to the two horizontal incisions, thereby reducing flap tension and widening the mastectomy field. In our case series, the NAC was grafted; indeed, it could be preserved if maintained with a bipedicle approach. The dermis of the lower pole is de-epithelialized and sutured to the pectoralis major muscle. In this technique, in contrast to the previous one, the incision axis is vertical, but the final skin reduction vector is horizontal, providing greater conization and thus projection of the breast.

### 2.2. Statistical Methods

Socio-demographic and clinical characteristics were analyzed using descriptive statistics. Categorical variables were reported with absolute frequencies and percentages and continuous variables with mean and standard deviations (SD). The Chi-squared test was used to compare the frequency of categorical variables between horizontal and vertical (inverted T) incision skin-reducing techniques, while the *t*-test was used to compare means of continuous variables.

All analyses were performed with SAS software v. 9.4 (SAS Institute, Cary, NC, USA).

## 3. Results

Twenty-four patients (30 breasts) were included in the study, eight underwent a skin-reducing technique with a horizontal incision and sixteen underwent a skin-reducing technique with an inverted T incision. Patients’ features are summarized in Table 1, while breast features are summarized in Table 2. All the socio-demographics and clinical characteristics were well balanced between the two groups, both for patients and breasts, except for NAC, which was treated with a pedicled approach in 80% of horizontal incisions compared to 100% with a grafted approach in the group of inverted T incision (*p*-value < 0.001). The mean satisfaction rating of patients was slightly higher in those who underwent the horizontal incision (Figure 3), except for the physical well-being and satisfaction with information, as shown in Table 3. All the outcomes at follow-up are shown in Table 4. After a median follow-up of 8 months (interquartile range, 7–14 months), out of eight patients who underwent the horizontal incision technique, none of them experienced any late complications and no patient underwent explantation of the prosthesis. Only one patient experienced an early flap necrosis. Out of the 16 patients who underwent the classic incision technique (Figure 4), 5 breasts had early complications, the most frequent being recurrent seroma (2 cases), followed by mastectomy flap necrosis (Figure 5), wounds dehiscence and nipple-areola complex necrosis in one case each. Only one patient underwent explantation of the implant due to wound dehiscence at the inverted T junction, followed by implant exposure. The remaining complications, including flap necrosis, wound dehiscence without implant exposure, and seroma formation, were all successfully managed conservatively in an outpatient setting, achieving complete resolution.

## 4. Discussion

The advancement and expanded implementation of breast screening have led to an upsurge in mastectomy procedures for breast cancer, subsequently elevating the demand for breast reconstructions. With the rise in such patients, challenges in reconstructing large breasts have become more apparent. The primary concerns in large breast reconstructions involve the management of excess post-mastectomy skin, compromised vascularization of the nipple–areola complex, and diminished blood supply to mastectomy flaps.

Over time, various reconstructive strategies have been put forth to surmount these issues. Notably, the SRM leveraging a wise pattern approach has been effective in mitigating excess skin, thereby enhancing the NAC position and volume reduction. This method has been shown to achieve symmetry with the contralateral breast, aptly correct breast ptosis, and deliver superior cosmetic outcomes with respect to projection and scar concealment. Nonetheless, despite iterative refinement, the technique does not fully eliminate the risk of vascular distress to flaps, which can precipitate implant exposure and failure of the reconstruction, resulting in delayed care and patient dissatisfaction.

Postoperative complications have been well-documented by several authors, including NAC ischemia or necrosis, distal mastectomy flap compromise, T-junction dehiscence [6], and impaired wound healing. For instance, Danker et al. [7] reported a 9.6% incidence of mastectomy flap necrosis after employing a wise pattern skin-reducing mastectomy with an inferior dermal flap for direct-to-implant reconstruction. Similarly, Bayram et al. [8] observed a 7.7% occurrence of partial NAC necrosis in comparable procedures.

Attributable predominantly to inadequate vascularization of the mastectomy flaps and NAC, these complications pose a significant concern. A mitigatory approach, introduced by Kontos [4], involves a horizontal incision that ostensibly preserves vascular integrity, thus reducing complication risks.

However, this technique may lead to more conspicuous scarring and challenges in achieving breast symmetry. Our study aimed to juxtapose the outcomes of classic skin reduction with this approach, comparing outcomes, complications, practical implications and degree of patient satisfaction.

The findings suggest that horizontal incision methods have potential benefits, such as sustaining robust NAC vascularization, which, however, does not necessarily translate to an improved quality of life compared to reinnervation. Of particular note is the absence of a T-junction, a common site of vascular compromise in wise pattern procedures [6], underpinning the superior safety profile of horizontal incision techniques. The enhanced vascular reliability frequently permits implant utilization, as the weight of the implant is not borne directly by the scar junction, and the skin flaps exhibit greater vascularization. These properties render the technique suitable in cases where the oncological disease is advanced in order to permit fast access to adjuvant therapies, such as radiotherapy since the complication rate is lower, providing immediate implant-based breast reconstruction. It should indeed be considered in cases where a higher risk of complications is present due to a fragile blood supply or the presence of additional risk factors, such as poor tissue quality or comorbidities such as diabetes mellitus. The use of acellular dermal matrix, specifically the ADM NATIVE type, increases the thickness of the lower flap and is adopted in cases of reduced flap thickness and to better fixate the inframammary fold [9]. Moreover, the ingenious application of ADMs, particularly when anchored to the inframammary fold, precludes excessive tension between this fold and the areola, significantly diminishing the likelihood of wound dehiscence, ischemia, and necrosis of the mastectomy flap [1].

The main drawback includes potentially visible scarring, which, in unilateral surgeries, can affect symmetry with the contralateral breast with inverted T procedures. In bilateral cases, however, maintaining the scars at a uniform height can enhance symmetry.

It should be added that the upper pole is generally less defined, and it is not possible to elevate the nipple–areola complex to 18–19 cm; otherwise, the folding of the de-epithelized flap is excessive with possible vascular compromise.

Indications of such a procedure are quintessential to obtain the best results and to make it comparable with the traditional inverted T procedure; in particular, we adopt such a procedure in moderate hypertrophic breast with no more than 26 cm of nipple to sternal notch distance.

This is due to the fact that the elevation of the NAC is limited to 6 cm; otherwise, the folding of the skin flap in the case of a higher length would cause nipple vascularity. Such limitation is not present in traditional skin-reducing techniques, which could be adopted in any type of breast ptosis.

Care should be adopted to avoid excessive elongation of the nipple-to-fold distance when planning the external incision, taking into consideration no more than 7 cm.

In our study population, we observed a reduced complication rate, both acute and late, in horizontal skin-reducing procedures when compared with inverted T procedures. According to BREAST-Q scores, we observed higher values in the horizontal group, although we noticed a reduced score of satisfaction with information sections, which stimulates us to improve the quality of information given to patients, especially in cases of such a technique that adopts specific scars.

While our findings provide valuable insights, we acknowledge the limitations of our study, particularly the small sample size and retrospective design. Further prospective studies with larger cohorts are warranted to validate these findings and provide more definitive recommendations regarding the optimal utilization of each technique.

This study is the first of its kind to compare these two techniques utilizing the BREAST-Q questionnaire, albeit with limitations such as its retrospective nature and a relatively small sample size for a conclusive assessment.

## 5. Conclusions

To our knowledge, the present article is the first to compare an inverted T traditional skin-reducing procedure with the Kontos technique. We are convinced that for moderately ptotic gigantomastic breasts, the horizontal incision skin-reducing technique could be a useful procedure to minimize the risk and obtain good results, especially in bilateral cases.

Information needs to be given to patients in order to let them comprehend the benefits of such an approach. Our work confirms the complexity of a modern breast reconstructive approach where several different procedures are available, and therefore, the optimal technique is one that is custom-tailored to the individual patient, carefully considering breast volume, tissue characteristics, personal risk factors, and preferences.

## Figures and Tables

**Figure 1 jcm-13-06276-f001:**
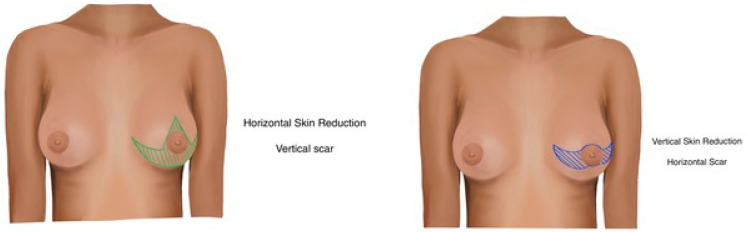
Vertical (inverted T) incision pattern with horizontal skin reduction (**Left**), horizontal incision pattern with vertical skin reduction (**Right**).

**Figure 2 jcm-13-06276-f002:**
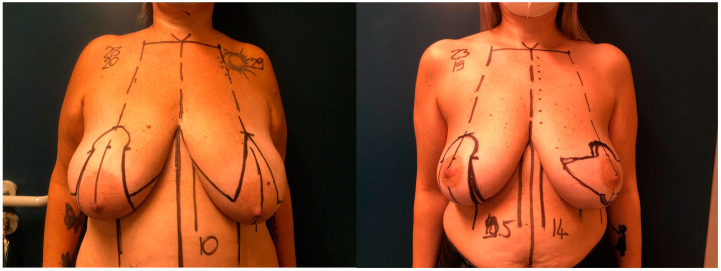
Pre-operative markings: vertical (inverted T) incision pattern with horizontal skin reduction on the left breast and wise pattern for contralateral breast reduction on the right breast (**Left**), Horizontal incision pattern with vertical skin reduction on the left breast and wise pattern for contralateral breast reduction on the right breast (**Right**).

**Figure 3 jcm-13-06276-f003:**
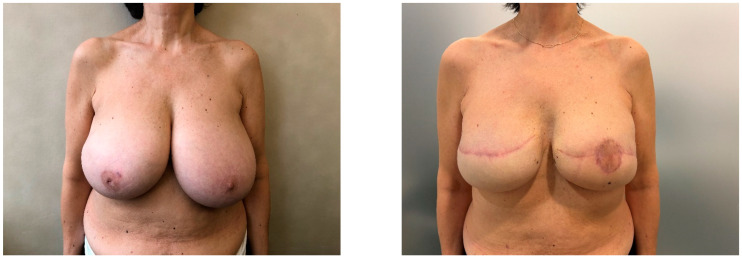
Patient before surgery (**Left**), same patient 3 months post-op: horizontal incision pattern with vertical bilateral skin reduction mastectomy and immediate implant-based reconstruction (**Right**).

**Figure 4 jcm-13-06276-f004:**
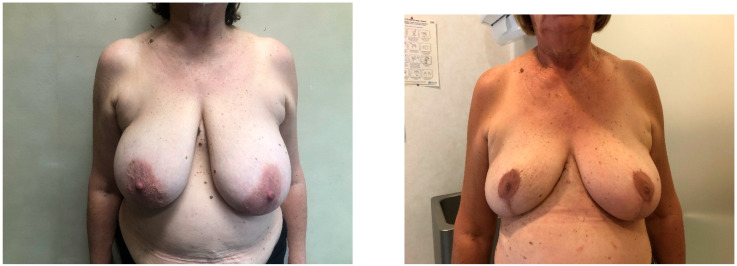
Patient before surgery (**Left**), same patient 3 months post-op: vertical incision pattern with horizontal bilateral skin reduction mastectomy and immediate implant-based reconstruction (**Right**).

**Figure 5 jcm-13-06276-f005:**
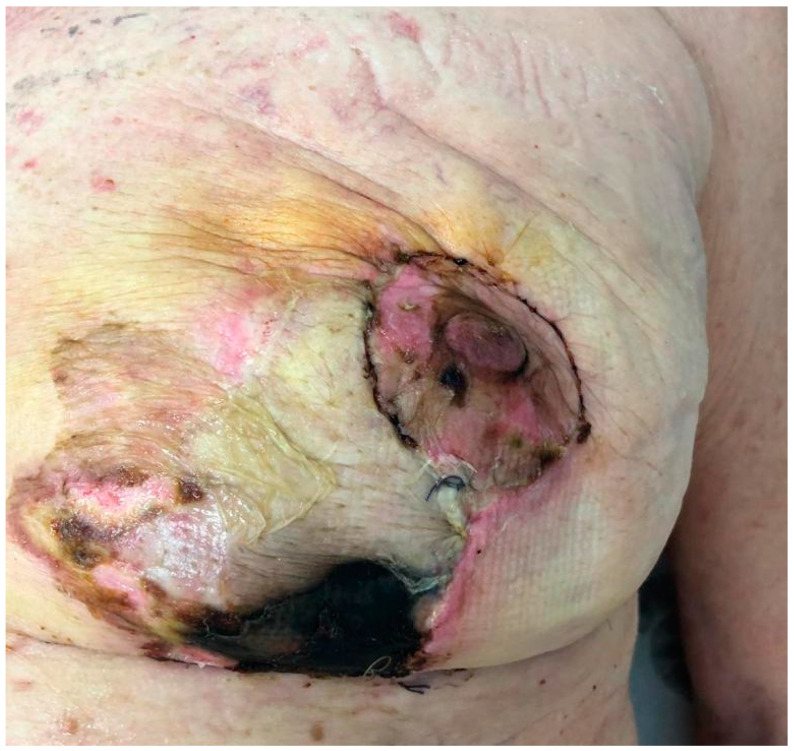
Early necrosis of the medial mastectomy flap after inverted T skin reduction technique, treated conservatively by toilette, debridement and multiple medications until complete healing.

**Table 1 jcm-13-06276-t001:** Patient characteristics (N = 24 patients).

	Skin Reducing Technique				*p*-Value	Overall (n = 24)	
	Horizontal (n = 8)		Vertical (Inverted T) (n = 16)				
	n	%	n	%		n	%
** *Matching variables* **							
**Age (years)**, mean (SD)	54.3 (6.9)		55.1 (9.2)		***	54.8 (8.4)	
**BMI**					***		
Normal weight (<25)	3	37.5	7	43.8		10	41.7
Overweight (25–30)	5	62.5	8	50.0		13	54.2
Obese (≥30)	0	-	1	6.3		1	4.2
Mean (SD)	23.9 (2.8)		25.1 (1.9)			24.7 (2.2)	
** *Other variables* **							
**Smoker**					*0.24*		
No	6	75.0	13	81.3		19	79.2
Yes	2	25.0	1	6.3		3	12.5
Unknown	0	-	2	12.5		2	8.3
**Comorbidities**					*0.75*		
No	6	75.0	11	68.8		17	70.8
Yes	2	25.0	5	31.3		7	29.2
*Asthma*	*0*		*1*			*1*	
*Type 2 diabetes mellitus and endometriosis*	*0*		*1*			*1*	
*Hypertension*	*1*		*0*			*1*	
*Hypothyroidism*	*0*		*1*			*1*	
*Hypothyroidism and epilepsy*	*0*		*1*			*1*	
*Hashimoto’s thyroiditis*	*1*		*1*			*2*	
**Genetic test performed**					*0.84*		
No	5	62.5	13	81.3		18	75.0
Yes	1	12.5	2	12.5		3	12.5
Unknown	2	25.0	1	6.3		3	12.5
**Neo-adjuvant chemotherapy**					*0.22*		
No	4	50.0	12	75.0		16	66.7
Yes	4	50.0	4	25.0		8	33.3
**Adjuvant chemotherapy**					*0.22*		
No	4	50.0	12	75.0		16	66.7
Yes	4	50.0	4	25.0		8	33.3
**Neo-adjuvant radiotherapy**					*-*		
No	8	100.0	16	100.0		24	100.0
Yes	0	-	0	-		0	-
**Adjuvant radiotherapy**					*0.44*		
No	6	75.0	14	87.5		20	83.3
Yes	2	25.0	2	12.5		4	16.7

* Matching variables.

**Table 2 jcm-13-06276-t002:** Breast characteristics (n = 30 breasts).

	Skin Reducing Technique				*p*-Value	Overall (n = 30)	
	Horizontal (n = 10)		Vertical (Inverted T) (n = 20)				
	n	%	n	%		n	%
** *Matching variable* **							
**Prosthesis volume (cc)**					***		
300–399	2	20.0	4	20.0		6	20.0
400–499	8	80.0	11	55.0		19	63.3
≥500	0	-	5	25.0		5	16.7
Mean (SD)	437.5 (60.2)		452.6 (57.2)			447.5 (57.6)	
** *Other variables* **							
**pT**					*0.95*		
In situ	1	10.0	2	10.0		3	10.0
0	0	-	1	5.0		1	3.3
1	4	40.0	8	40.0		12	40.0
2	4	40.0	8	40.0		12	40.0
X	1	10.0	1	5.0		2	6.7
**pN**					*0.58*		
0	6	60.0	13	65.0		19	63.3
1	2	20.0	5	25.0		7	23.3
2	0	-	1	5.0		1	3.3
3	1	10.0	0	-		1	3.3
X	1	10.0	1	5.0		2	6.7
**Subtype**					*0.22*		
HR+	4	40.0	13	65.0		17	56.7
HER2+	2	20.0	5	25.0		7	23.3
Triple negative	3	30.0	2	10.0		5	16.7
X	1	10.0	0	-		1	3.3
**Axillary dissection**					*0.33*		
No	7	70.0	17	85.0		24	80.0
Yes	3	30.0	3	15.0		6	20.0
**Nipple-Areola Complex**					*<0.001*		
Grafted	2	20.0	20	100.0		22	73.3
Pedicled	8	80.0	0	-		8	26.7
**Prosthesis type**					*0.44*		
CPG 311	1	10.0	3	15.0		4	13.3
CPG 312	3	30.0	11	55.0		14	46.7
CPG 313	5	50.0	4	20.0		9	30.0
CPG 322	0	-	1	5.0		1	3.3
MENTOR MH	1	10.0	1	5.0		2	6.7
**Number of days of drainage**, mean (SD)	13.7 (3.4)		14.2 (3.2)		*0.70*	14.0 (3.2)	

* Matching variables.

**Table 3 jcm-13-06276-t003:** BREAST-Q questionnaire (n = 24 patients).

	Skin Reducing Technique				*p*-Value
	Horizontal (n = 8)		Vertical (Inverted T) (n = 16)		
	Mean	SD	Mean	SD	
Satisfaction with breast	72.5	10.8	65.1	20.0	*0.34*
Satisfaction with implant	74.8	28.6	65.5	19.1	*0.35*
Satisfaction with results	93.3	5.0	89.4	10.2	*0.33*
Psychosocial well-being	81.0	11.8	70.3	26.4	*0.18*
Sexual well-being *	70.5	16.5	57.4	33.1	*0.31*
Physical well-being: chest	37.3	12.6	49.9	20.0	*0.12*
Satisfaction with information	59.0	7.0	65.8	16.8	*0.18*
Satisfaction with surgeon	95.8	4.9	82.4	19.6	*0.02*
Satisfaction with medical team	100.0	0	85.8	10.2	*<0.001*
Satisfaction with office staff	99.0	1.9	85.0	18.3	*0.008*

* N = 14 for vertical group.

**Table 4 jcm-13-06276-t004:** Outcomes at follow-up (N = 30 breasts).

	Skin Reducing Technique				*p*-Value	Overall (n = 30)	
	Horizontal (n = 10)		Vertical (Inverted T) (n = 20)				
	n	%	n	%		n	%
**Early complications**					*0.33*		
No	9	90.0	15	75.0		24	80.0
Yes	1	10.0	5	25.0		6	20.0
*NAC necrosis*	*0*		*1*			*1*	
*Dehiscence*	*0*		*1*			*1*	
*Flap necrosis*	*1*		*1*			*2*	
*Sieroma*	*0*		*2*			*2*	
**Late complications**					*0.47*		
No	10	100.0	19	95.0		29	96.7
Yes	0	-	1	5.0		1	3.3
*Sieroma*	*0*		*1*			*1*	
**Breast explant**					*0.47*		
No	10	100.0	19	95.0		29	96.7
Yes	0	-	1	5.0		1	3.3

## Data Availability

The raw data supporting the conclusions of this article will be made available by the authors on request.

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
