# Peer review of "Comparative Efficacy of Classic Versus Horizontal Incision Techniques in Skin-Reducing Mastectomy: A Single Center Retrospective Analysis"

_jcm, 2024, doi:10.3390/jcm13206276_

Round 1
Reviewer 1 Report
Comments and Suggestions for Authors
Even though, this is an interesting, and very well conducted study presented by the authors, and gives a hint of a potential benefit of the horizontal scar as described by Konto et al. in comparison to the traditional vertical scar method, the final statement, states that after all it will still be an individual decision not clearly advocating for one or the other technique.
I also would suggest the authors to distinguish between diabetes mellitus and endometriosis (Table 1 - Comorbidities).
Comments on the Quality of English LanguageIn Table 4 the term "Brest explant" should be replaced by "breast implant removal/ explantation"
Reviewer 2 Report
Comments and Suggestions for Authors
Dear authors
Your manuscript is very interesting, but the following aspects need to be improved:
1. Enhance the introduction to clearly state the research objective and emphasize the impact of skin-reducing mastectomy techniques in the existing literature. Highlight the need for further study in this area to set the stage for the manuscript.
2. Work on improving the transitions between sections to ensure a more logical and coherent flow throughout the manuscript.
4. Improve the quality of the pictures and add more images from additional cases to provide a comprehensive representation.
5. Ensure that the paper explicitly links results to practical implications for reconstructive plastic surgery. Providing specific examples that highlight the relevance of the findings to clinical practice would significantly enhance the paper.
6. Present the findings with a greater level of specificity, integrating specific examples that underscore the clinical relevance of the research to the field of plastic surgery.
7. Provide more definitive conclusions to wrap up the manuscript effectively a more explicit delineation of their practical implications for reconstructive plastic surgery. There is a need to present the findings with a greater level of specificity and instead integrating specific examples that underscore the clinical relevance of the research of breast reconstructive surgery. Moreover, the manuscript would benefit from more definitive conclusions.
Reviewer 3 Report
Comments and Suggestions for Authors
Overall, the study is well-written and engaging. However, some aspects need to be revised.
More studies in the literature should be discussed, the discussion section expanded, and more studies cited throughout the paper.
The limitations should be discussed further, as they are only briefly noted in the current version of the paper.
Comments on the Quality of English LanguageSome minor mistakes in the English language (syntax, grammar, etc.) need to be corrected throughout the paper.
For example, in the conclusion section, "We are convinced.... ....to minimize the risk and obtain good results, especially in bilateral cases." Minimize the risk for what? Please thoroughly revise the paper to correct such mistakes.
Reviewer 4 Report
Comments and Suggestions for Authors
Dear Editor and Authors,
Thank you for the opportunity to review the manuscript entitled “Comparative Efficacy of Classic versus Horizontal Incision Techniques in Skin-Reducing Mastectomy: A Single Center Retrospective Analysis”. The authors compared two techniques of skin reducing mastectomies —the horizontal incision and the classic inverted T incision—by examining their clinical and surgical outcomes. They analyzed data from 24 patients (30 breasts) who underwent SRM with immediate prosthetic reconstruction between 2019 and 2023. They concluded that horizontal incision technique was associated with enhanced patient satisfaction and reduced complication rates. They claim that the selection of the technique should be customized based on the patient's individual risk factors, tissue quality, and preferences. I consider the paper appropriate for the Journal /especially for oncologic/breast surgeons/ and worth publishing in this Journal.
However, I have some remarks:
- Include conclusions /not discussion/ in the abstract
- Material and methods – pls describe according to which criteria you qualified a patient for “only horizontal” vs. “classic” skin reduction /randomized? specific indications for each technique?is it related to the weight of resected tissues – please provide these data for both groups /and add a follow-up period you aimed for /in inclusion criteria/
- Perioperative photos /e.g. with a preoperative markings/ would be helpful to understand your technique and planning – I would value this more than the schematic figure you provided…
- Is caption for figure 1 correct?
- How you managed the described complications? Surgical revisions? Conservatively? How long it took you to manage comoplications? This should be described.
- For your example patients pls describe if nipples were grafter or transferred via pedicles /which pedicles?/
Round 2
Reviewer 4 Report
Comments and Suggestions for Authors
Dear Editor and Authors,
Thank you for including the requested information, I still have some remarks:
-for figure 5 pls provide a photograph with the final result /after conservative treatment/
- I suggest presenting your criteria of qualification to different skin reducing techniques in Material and methods /not in the conclusions/ as they are not based on your analysis, but rather on your clinical observations and experience
- still, I think that terms you are using “horizontal skin reduction” with vertical scar vs “vertical skin reduction” with horizontal scar are confusing! Pls use terms that all Readers can understand: Wise or inverted T pattern /as far as I am concerned it reduces skin in both vertical and horizontal” aspects…/
Author Response
- Comment (1): for figure 5 pls provide a photograph with the final result /after conservative treatment
- Response (1): unfortunately we do not own any photograph of the final result after conservative treatment of the complication in figure 5.
- Comment (2): I suggest presenting your criteria of qualification to different skin reducing techniques in Material and methods /not in the conclusions/ as they are not based on your analysis, but rather on your clinical observations and experience
- Response (2): Thank you for your suggestion and collaboration. We proceeded to present our criteria of qualifications to different skin reducing techniques in the section “Material and Methods” instead of “conclusions”
- Comment (3): still, I think that terms you are using “horizontal skin reduction” with vertical scar vs “vertical skin reduction” with horizontal scar are confusing! Pls use terms that all Readers can understand: Wise or inverted T pattern /as far as I am concerned it reduces skin in both vertical and horizontal” aspects…/
- Response (3): we proceeded to use different terms to define each technique. Vertical Skin Reduction Technique was changed to “Inverted T Technique” and Horizontal Skin Reduction Technique was changed to “Horizontal Incision Technique”, excluding any confusing term.
Thank you for your collaboration and suggestions to improve our work.